# Shaping Public Servant Well-Being: Lessons from Brazil’s SIASS Program

**DOI:** 10.3390/ijerph21101341

**Published:** 2024-10-10

**Authors:** Marcos Massaki Imamura, Gabrielly de Queiroz Pereira, José Roberto Herrera Cantorani, Luiz Alberto Pilatti

**Affiliations:** 1Electrical and Computer Engineering Department, Federal University of Technology—Parana, Campus Londrina, Londrina 86036-370, Brazil; imamura@utfpr.edu.br; 2Program in Electrical and Computer Engineering, Federal University of Technology—Parana, Campus Curitiba, Curitiba 80230-901, Brazil; 3Physical Education Department, Federal Institute of Education, Science and Technology of São Paulo, Campus Registro, Registro 11900-000, Brazil; cantorani@ifsp.edu.br; 4Production Engineering Department, Federal University of Technology—Parana, Campus Ponta Grossa, Ponta Grossa 84017-220, Brazil; lapilatti@utfpr.edu.br

**Keywords:** health, health actions, work safety, social assistance, public servants, federal university, Brazil

## Abstract

This study aims to identify how the health and well-being of public servants are promoted and managed at the Federal University of Technology—Paraná (UTFPR). The Integrated Subsystem for Attention to the Health of Public Servants (SIASS), established by the federal government in 2009, is an initiative that integrates health actions, work safety, and social assistance. This qualitative study utilized the analysis of institutional documents and semi-structured interviews with managers involved in SIASS implementation at the UTFPR. The analysis reveals that, despite the creation of the SIASS, the UTFPR faces challenges such as resource scarcity, reliance on procurement processes, and a reactive rather than preventive approach to occupational health issues. The structural diversity across campuses and the lack of systematic evaluation of working conditions further limit the effectiveness of policies. It is recommended that the UTFPR adopt a more integrated, data-driven, and proactive approach to occupational health management. Strengthening interinstitutional partnerships, optimizing resource allocation, and developing continuous evaluation systems are essential steps to ensure a work environment that effectively promotes the health and well-being of its employees.

## 1. Introduction

The health and well-being of public servants are essential for the proper functioning of public institutions, directly influencing the quality of services provided to the population [1]. A work environment promoting employees’ health improves public service quality and reduces absenteeism, occupational diseases, and burnout [2,3].

In Brazil, the general perception of public services is often negative, marked by high salaries and privileges compared to the private sector [4,5]. Initiatives such as the Integrated Subsystem of Attention to the Health of Public Servants (SIASS), which have little equivalence in the private sector and aim to promote well-being and better working conditions for public servants, are often seen as part of this privilege. However, when functional, these policies help to ensure the efficiency and quality of services provided to the population, especially considering health and occupational safety challenges.

Regarding the perception of the services provided, Brazilian public universities are an exception within the public sector. They are known for their academic excellence, frequently surpassing private institutions in national and international evaluations, and consistently occupying top global rankings [6]. Although the federal higher education system offers relatively high salaries and better research conditions than private institutions, it faces significant challenges due to the pressure on staff, limited resources, and increasing workload. These factors reinforce the need for effective occupational health policies. In this context, the SIASS can be a critical factor in ensuring a healthier and safer work environment for staff at federal institutions.

### 1.1. Overview of SIASS

The trajectory of health and well-being policies for workers in Brazil began to take shape late with the Consolidation of Labor Laws (CLT) in 1943, which guaranteed fundamental rights such as an eight-hour workday and paid vacations. However, these protections were initially directed toward the private sector, leaving public servants needing a specific occupational health framework. With the enactment of the 1988 Constitution [7], labor and social security rights were expanded, and these protections were extended to federal public servants through the Unified Legal Regime (RJU). The RJU implemented the first regulations focusing on employee health, including medical leave and disability pensions. Additionally, the creation of the Unified Health System (SUS) expanded access to health services, promoting universal care, including for public servants.

The evolution of these policies continued in the 2000s, with the establishment of more specific systems aimed at occupational health for federal public servants. The Integrated Subsystem for the Health Care of Public Servants (SIASS), established in 2009, marked a significant advance in occupational health management in the federal public service.

The federal government created the SIASS to integrate health, workplace safety, and social assistance initiatives for federal public servants in response to the growing demand for public policies that promote public servants’ health in a coordinated and systematic manner [8]. The subsystem was designed to centralize and standardize health promotion, occupational risk prevention, and medical follow-up procedures, and to facilitate access to treatments and prevention programs [9]. Its main objective is to adjust these occupational health practices to the specific needs of public servants, who perform essential functions for the functioning of the State. By conducting inspections, studying working conditions, implementing preventive measures, and educating employees on health and safety standards, the SIASS aims to ensure that federal public servants can work in a safer and healthier environment. Additionally, it promotes negotiations with managers to improve work environments.

However, despite its ambitious goals, implementing the SIASS has faced several practical challenges over the years. The federal institutions covered by the legislation exhibit distinct characteristics that affect the execution of SIASS policies. Some of the main challenges include disparities in resources allocated to occupational health, a shortage of qualified personnel, and bureaucratic constraints, which compromise the program’s effectiveness [10,11]. In some institutions, SIASS was implemented late; in others, the need for more professionals and the absence of autonomy to expand staff numbers further aggravated the situation [12].

Furthermore, the uneven execution of the program across different institutions reflects the complexity of Brazil’s public occupational health system. Although the SIASS was designed to standardize health practices across federal institutions, these practices often vary considerably depending on the availability of resources and local organizational structures [13]. Some federal universities, particularly those without medical or related courses, such as the Federal University of Technology—Paraná (UTFPR), the institution examined in this study, face specific challenges. The absence of health-related courses and the shortage of specialized professionals significantly hinders the implementation of an effective occupational health program.

In the context of this study, understanding how the SIASS is applied at the UTFPR allows for assessing the progress made and identifying the obstacles that still need to be overcome. These challenges must be addressed to ensure public servants receive the necessary support to maintain their health and well-being. This analysis allows for evaluating the direct impact of health and well-being policies on employees’ quality of life and, consequently, on the effectiveness of institutional activities.

This study not only aims to explore the challenges and opportunities of SIASS implementation, but also aims to provide a foundation for improving occupational health policies aimed at public service in Brazil, offering contributions to global discussions on managing the health and well-being of public servants in similar contexts worldwide.

### 1.2. Derivation of the Research Questions

Previous studies have explored the implementation of the SIASS in various contexts in Brazil, highlighting its institutionalization in Rio de Janeiro [14] and its application in providing healthcare to Federal Institute employees [15]. Research has also pointed to the SIASS as an example of collaborative efforts to promote sustainability in higher education institutions (HEIs) [16]. However, the inconsistent implementation across different regions and institutions has led some researchers, such as Moura [17], to question its effectiveness as a public policy. These studies suggest that although the SIASS offers significant potential, there are still many gaps in the literature regarding its comprehensive evaluation.

Globally, similar challenges have been observed in the public sector of other countries. Kilpatrick et al. [18,19] described how employees in Australia align their health goals with their needs, while Marcatto et al. [20] highlighted the impact of excessive workloads on public employees in Italy. In Pakistan, Hassan et al. [21] explored the link between well-being and motivation in the public sector, and Adamopoulos and Syrou [22] identified occupational hazards in Greece. Studies in the USA conducted by Leider et al. [23] examined turnover trends, while research in Denmark emphasized the effect of social capital on absenteeism [24]. Internationally, these studies point to common issues in managing employee health and well-being, reinforcing the need for a policy framework that is more aligned with workers’ demands, as explored in the case of Brazil.

Given the current state of SIASS implementation and the challenges observed in promoting the well-being of public servants, this study addresses the following research questions:How are health and well-being managed and promoted at a federal university with unique characteristics, such as the absence of a medical school?What are the impacts of limited discretionary resources and lack of autonomy in staff expansion on implementing the SIASS in these institutions?How can the lessons from this study contribute to the global debate on workplace health practices in similar contexts?

This study aims to explore the challenges and opportunities of SIASS implementation and provide a basis for improving occupational health policies aimed at public service in Brazil.

## 2. Materials and Methods

This study adopts a qualitative approach, focused on documental research, to analyze the health and well-being policies implemented at the UTFPR. Documental research is a methodological procedure that uses primary sources, such as reports and regulations, for data collection and is especially effective when seeking to understand institutional practices in a historical and normative context [25].

### 2.1. Context and Rationale

The UTFPR was chosen as the focus of this study due to its specific and unique characteristics within the federal higher education system. The institution stands out as Brazil’s only technological university. It focuses on engineering courses, differentiating it from federal universities with medical schools and Federal Institutes, which focus on professional and technological education. This characteristic was considered essential for the study, as it allowed an analysis of how health and well-being policies are implemented in an environment where resources and expertise in occupational health are not directly linked to an academic structure focused on health, as is the case in universities with medical programs.

This study did not consider other institutions, including universities with medical schools, because the goal was to analyze a context closer to the reality of most federal institutions. Universities with medical schools represent an exception in the federal higher education system. Therefore, the UTFPR was chosen to ensure the relevance of the findings for most institutions. The research did not aim to compare different types of universities, but to provide an in-depth analysis of an institution without medical infrastructure.

This methodological choice was deliberate to explore the health and well-being policies in an institution whose structure and challenges resemble most federal Brazilian universities, which also do not have medical programs.

### 2.2. Data Collection

Data were collected through an extensive documental analysis covering institutional documents produced between 2009 and 2022 (Table 1). This period was chosen to provide a comprehensive view of the evolution of health and well-being policies since the creation of the SIASS.

The types of documents analyzed were
Annual Management Reports: These documents detail the strategic goals, resource allocation, and challenges faced by the UTFPR in implementing its health and well-being policies. They are essential for tracking the development of institutional practices;Internal regulations and institutional norms: These documents were analyzed to understand the regulatory framework guiding occupational health and safety policies within the UTFPR;Relevant federal legislation and guidelines: This analysis included documents related to the SIASS, which establishes guidelines for health policies for federal public servants. These documents allow verification of the alignment of the UTFPR’s practices with federal legal requirements.

The analysis of these documents was conducted systematically. All the relevant documents were analyzed to ensure a broad view of the implemented policies without limiting the selection to the most important documents. The documental approach was the primary data source for the study.

Additionally, semi-structured interviews were conducted with two managers directly involved in implementing the SIASS (coded as E1 and E2) to complement the documental analysis. The interviews provided more detailed information on the practical application of the documented policies and the specific challenges faced in managing health and well-being. These interviews were not the primary basis of the study, but were used to enrich the documental analysis, offering an additional perspective on institutional practices. The interviews lasted an average of 38 min, and were recorded and later transcribed using NVivo 12 Pro software, with the transcripts reviewed by the interviewees to ensure the accuracy of the information.

### 2.3. Data Analysis

Data analysis was mainly based on documental review, complemented by information obtained from the interviews. The documents were systematically analyzed, following the steps recommended for documental analysis [26]. The process involved five phases:Compilation: All the documents were organized into a structured database, grouping information from different sources, such as management reports, internal regulations, and federal guidelines;Deconstruction: The documental data were divided into smaller parts, identifying the main themes and patterns present in the texts;Reassembly: The data were then coded and reorganized into main categories, such as Inspection and Prevention, Investigation and Follow-up, Health Education and Promotion, and Negotiation and Improvement, aligned with the research questions;Interpretation: The comparative analysis of the documents identified gaps between the formally established policies and their practical application. The information obtained was compared with the existing literature, generating an understanding of the effectiveness of health and well-being policies at the UTFPR;Conclusion: The final synthesis emphasized the practical implications of the policies, highlighting both the advances and the areas that need improvement.

In addition to the documental analysis, the semi-structured interviews were subjected to rigorous analysis. After being transcribed using NVivo software, the interviews were analyzed using an open coding process. This process involved identifying meaning units from the managers’ statements related to the challenges and implementation of health and well-being policies at the UTFPR.

The coding followed these steps:Initial coding: The transcripts were reviewed, and relevant excerpts were assigned preliminary codes, focusing on the core themes of the research, such as the implementation difficulties, the perceptions of the SIASS, and the occupational health promotion practices;Categorization: The codes were grouped into broader categories that reflected institutional barriers and strategies to overcome these challenges. The main categories included Operational Challenges, Policy Implementation, Impact on Working Conditions, and Resource Allocation;Interpretation: After categorization, the data were interpreted in terms of the institutional and normative context provided by the documental analysis. This triangulation allowed the comparison of the managers’ perceptions with the analyzed documents, identifying consistency and discrepancies between institutional discourse and practice;Integration: The results of the interview analysis were integrated with the documental data, enriching the understanding of the analyzed policies and providing a complete perspective on the challenges and opportunities faced in implementing health and well-being policies.

The interview analysis served as a fundamental complement to the documental analysis, allowing a practical and in-depth view of health and well-being policies from the experience of the managers involved in their implementation.

### 2.4. Modeling the Report Structure

The modeling chosen for the article follows the operationalization of the IMRaD structure. It is worth noting that adherence to this model is due to its simplicity and its logical form of presenting the research results, which allows for more palatable reading of the research elements [27].

### 2.5. Ethical Considerations

Although this study was predominantly based on public institutional documents, semi-structured interviews were also conducted during the data collection. All interview participants provided informed consent before their inclusion in this study. The research adhered to the principles of the Helsinki Declaration, ensuring the protection of the rights and privacy of the interviewees. The research protocol was approved on 1 August 2024 by the Ethics Committee of the UTFPR, Campus Medianeira (Project ID: 6.977.762).

## 3. Results

The management reports of the UTFPR emphasize workplace health and well-being policies and practices. Table 1 synthesizes this data, offering a panoramic view of institutional trends and priorities in this area.

**Table 1 ijerph-21-01341-t001:** Summary of health and well-being actions in annual reports.

Year	Health and Well-Being in Annual Reports
2009	The report emphasizes the importance of promoting employees’ quality of life and well-being, including performance evaluation and strategic objectives to improve educational and administrative environments.
2010	The institution’s commitment to health and well-being in the workplace stands out, with health programs, a socialized health plan, and organizational climate surveys.
2011/2012	There is no specific information on health and well-being in the workplace.
2013	Concerns about replacing employees and mentions of absences due to health reasons exist.
2014	Projects and works aim to improve the quality of environments, focusing on ergonomic and thermal comfort.
2015	Emphasis is placed on training and professional development policies, focusing on physical infrastructure.
2016	Creating an anti-moral harassment policy and health and quality of life programs stand out.
2017	The highlight is the Health Plan offered to employees and their dependents.
2018	Advances in structuring the SIASS/UTFPR/IFPR and training actions are mentioned.
2019	Improvements to the organizational climate assessment process and investments in projects related to mental health are highlighted.
2020	Emphasis on the importance of health and well-being, especially during the COVID-19 pandemic, with sanitary measures and solidarity actions.
2021	Highlights include health promotion and awareness-raising actions, such as live broadcasts.
2022	Approaches to actions to improve employees’ quality of life, with specific investments in mental health and proportional decentralization across the institution’s campuses.

Source: UTFPR Management Reports (2009–2022) [28,29,30,31,32,33,34,35,36,37,38,39,40,41].

The reports do not disclose the financial amounts allocated to health and well-being initiatives, except for the Management Report related to the 2022 fiscal year. This report highlighted investments in mental health and workplace quality of life, totaling BRL 82,743.29 (approximately USD 16,500.00). This investment represents approximately 0.0075% of the UTFPR’s total budget for the 2022 fiscal year. According to interviews with key informants (Interviewees 1 and 2) [42,43], financial allocations for health and well-being have historically been minimal due to budget constraints, competing priorities across different management areas, and legal limitations. This trend reflects a preference for allocating resources to core mission areas at the expense of employee well-being initiatives.

Despite identifying various health and well-being actions in the annual reports, a deeper analysis reveals several challenges. The identified actions are often localized and associated with minimal financial allocations, as the 2022 budget data indicates. Institutional reports do not present detailed results on the effectiveness of these initiatives, limiting the ability to assess the real impact of these actions over time. Additionally, the interviewees did not have detailed information on the results of these measures. Their responses indicated that the reach and effectiveness of the initiatives varied and generally did not conform to a long-lasting institutional policy. This variability highlights the need for more structured and transparent monitoring of the impact of health and well-being measures across the institution. The interviewees emphasized that, while important, the initiatives often needed more institutional support to become fully integrated and effective programs.

The reference year chosen to illustrate the university’s health-related activities was 2019. During this period, the UTFPR’s SIASS operated smoothly, unaffected by the subsequent impacts of the COVID-19 pandemic. This year is a representative example of the functioning and challenges the university’s employee healthcare systems face under normal circumstances.

In 2019, the SIASS supported UTFPR employees by conducting approximately 600 medical assessments. These assessments included evaluations for personal health treatment leaves, family-related illness leaves, and other health-related assessments. However, around 250 assessments could not be completed due to a shortage of physicians at specific campuses and government-imposed hiring restrictions during that year. To address these challenges and ensure comprehensive care, SIASS–UTFPR collaborated with SIASS–IFPR (the health care system of the Federal Institute of Paraná) to initiate the accreditation process for medical experts. This effort aimed to guarantee that all UTFPR employees had access to necessary medical assessments, aligning with legal requirements and facilitating an understanding of the health issues employees face and the implementation of appropriate health programs.

In addition to SIASS activities, the university’s occupational safety sector conducted technical assessments, evaluated workplace environmental conditions, provided technical guidance, and issued PPP forms. The PPP (Profile and Environmental Risk Factor Document) is a Brazilian document that details workers’ exposure to hazardous agents and special working conditions for retirement purposes. The sector also engaged in preventive actions against workplace accidents, investigated and documented incidents, and provided technical reports on environmental working conditions. The Employee Welfare Department (SEBEN) was critical in supporting employee quality of life, organizing initiatives such as flu vaccination campaigns, health and dental insurance plans, and the Integrated Community in the Multiplication of Knowledge (CIMCO) program.

The national dental plan provided by the UTFPR encompasses a wide range of accessible dental treatments for active employees, retirees, their dependents, and pensioners. Simultaneously, the university’s health plan serves as a cornerstone in promoting the well-being of its workforce, granting access to quality healthcare services.

In Brazil, the quality of services offered by SUS (the national public health system) is widely questioned. Individuals with higher purchasing power often opt for private health and dental plans. The UTFPR manages these plans, offering them to employees at more affordable rates without providing subsidies due to Brazilian legislative constraints. Those who choose private plans receive partial reimbursement from the federal government.

Table 2 summarizes the responses obtained during interviews with UTFPR managers regarding health and well-being practices and challenges. The table highlights critical areas related to inspection and prevention, investigation and follow-up, health education, and promotion, negotiation, and improvement. Additionally, it sheds light on the impact of not having a medical program within the institution.

The information gathered from the interviewees highlights the challenges in executing health and well-being initiatives at the UTFPR. The allocation of discretionary resources is limited and primarily focuses on legal compliance and risk mitigation, with insufficient emphasis on preventive and proactive measures. The initiatives identified are predominantly reactive and aimed at fulfilling regulatory obligations rather than fostering strategic and continuous improvements in the staff’s quality of life.

The interviewed managers noted that while initiatives to enhance employee well-being do exist, they are often limited in number and tend to be short-lived. Most actions respond to immediate demands without a sustained plan for long-term improvements. Despite budgetary constraints, occasional efforts to address emerging needs were mentioned; however, the persistent lack of resources and specialized personnel continues to hinder the development of broader and more effective programs.

## 4. Discussion

Document analysis and interviews were essential to understand the scenario of the implementation of occupational health policies in the Brazilian public sector. These two data sources allow us to identify advances related to the standards that govern the health and well-being of federal public servants and the challenges for fully implementing these policies in federal institutions.

### 4.1. Document Analysis

Brazilian legislation establishes that the health and well-being of federal public servants are priorities, with comprehensive policies involving prevention, health assistance, and workplace safety. Specifically for public servants, regulations such as the RJU and programs like the SIASS were created to promote these policies, offering a complete approach to occupational health and ensuring a healthy and safe environment for public employees. However, SIASS implementation faces significant challenges in many institutions, particularly regarding the lack of financial and human resources. Document analysis reveals that the budget allocated to universities by the federal government is often insufficient to meet all needs, resulting in a fragmented and frequently reactive implementation of the SIASS at the UTFPR. The lack of a specific budget forecast to meet the SIASS requirements, such as hiring specialized professionals, is one of the main obstacles that compromise the complete execution of health and safety policies [17].

Institutional documentation shows that the UTFPR conducts technical evaluations of work environments with the participation of safety engineers and contracted companies. However, these actions are limited by insufficient staff, delaying coverage across all campuses and compromising the effectiveness of preventive measures. Additionally, reliance on bidding processes to hire third-party companies often delays implementing safety measures, as highlighted in the management reports.

### 4.2. Interview Analysis

The interviews with the UTFPR managers provided a more detailed overview of the practical challenges in implementing health and safety policies. Both managers consistently highlighted the shortage of human and financial resources as the main obstacle. Manager E1 emphasized that “we only have one safety engineer to cover all campuses” [42], which compromises the ability to conduct regular inspections and implement preventive measures promptly in a large institution. This point confirms the findings of the document analysis, which indicate a lack of specialized personnel.

Moreover, the interviews revealed that the actions implemented at the UTFPR tend to be predominantly reactive. Manager E2 observed that “health and safety actions at work are largely a response to specific demands, rather than following preventive planning” [43]. This reactive approach limits the long-term impact of occupational health policies, which should be aligned with SIASS guidelines, promoting continuous and preventive health measures [15].

Another point highlighted in the interviews was the bureaucracy involved in bidding processes for contracting outsourced services and, within these, the difficulty in achieving the expected results, which delays or hinders the implementation of corrective and preventive measures. Manager E1 mentioned that “the reliance on bidding processes can result in significant delays, compromising the effectiveness of preventive actions” [42]. This factor is widely discussed in the literature as one of the main obstacles to the effective implementation of public health policies in occupational health. For Martins et al. [14] and Mendonça et al. [15], bureaucracy and the lack of autonomy in hiring are critical barriers to the efficient implementation of health policies in the public sector.

Another critical point raised in the interviews was the need for a formal system for evaluating health promotion campaigns. Manager E2 admitted that “there is no systematic evaluation of campaign results, which makes it difficult to adapt strategies to the needs of the servers” [43]. This fact reflects a gap in institutional management that directly affects the effectiveness of the implemented policies.

Finally, both managers converge on critical points, such as creating this and other laws without adequate conditions for complete execution. Bureaucracy and legal impediments render the implementation process of occupational health policies unfeasible, culminating in non-compliance with laws. Furthermore, although prioritizing finalist areas (mainly teaching and research) is somewhat justifiable, it reveals an imbalance by relegating health and safety issues to a secondary position, compromising the well-being of the staff and the sustainability of long-term policies.

### 4.3. Comparison between Document Analysis and Interviews

The comparison between document analysis and interviews reveals significant consistencies between the theoretical and practical challenges the UTFPR faces. No inconsistencies were found between the interview data and the analyzed documents. Data triangulation shows that omissions in institutional reports—a kind of “silence” on specific topics—reflect the reactive practices described by the interviewed managers. This convergence between document analysis and interviews reinforces the identification that the UTFPR health and safety actions are implemented without priority and adequate preventive planning.

The predefined categories—Inspection and Prevention, Investigation and Monitoring, Health Education and Promotion, and Negotiation and Improvement—were derived from relevant legislation regulating occupational health in public institutions, precisely the SIASS program guidelines. These categories were used to guide the interviews and document analysis. The interviews provided practical information about the real-world application of these categories. At the same time, the institutional documentation was systematically analyzed to ensure comprehensive coverage of the UTFPR’s health and well-being policies.

Both data sets indicate that the lack of financial and human resources and excessive bureaucracy compromise the institution’s ability to implement occupational health policies effectively. The document analysis provides a broader view of institutional challenges, such as the lack of budget forecasting and the reliance on bidding processes. At the same time, the interviews offer a detailed account of the difficulties managers face in the day-to-day execution of these policies. These sources reveal that the UTFPR’s policies tend to reflect reactive measures, when possible, rather than proactive and preventive strategies. Both levels of analysis reinforce the need for a more integrated and proactive approach that is not limited to reactive responses to health and safety emergencies.

Additionally, institutional documents enabled data triangulation, particularly in identifying how these predefined categories are reflected in the university’s formal practices. While the interviews provided firsthand perspectives, document analysis was crucial in verifying and cross-referencing these categories with the UTFPR’s institutional practices over time. This approach ensured that both sources supported the conclusions, providing a more robust understanding of health management policies at the UTFPR.

Furthermore, the absence of a medical school at the UTFPR presents additional challenges to implementing occupational health policies. Without a medical school, the institution’s ability to develop integrated programs internally is limited, increasing its reliance on external services. This fact hinders prioritizing effective and comprehensive health measures for its staff, making advancing well-being in the university environment more challenging.

Although one of the interviewees stated that the absence of a medical school at the UTFPR would not significantly impact occupational health policies, it is essential to consider both practical and ideal scenarios. The interviewee was correct in suggesting that, in an ideal scenario, health professionals assigned to the SIASS would ensure the effectiveness of policies, regardless of the existence of a medical school. In that case, the dedicated resources and personnel would be sufficient to provide occupational health services.

However, in practice, universities with medical schools often involve students and faculty in activities beyond teaching, offering additional support to the health needs of the academic community through partnerships, extension programs, or clinics that integrate teaching and healthcare delivery. As a result, these universities can implement more integrated and comprehensive occupational health programs internally, reducing their reliance on external providers.

Thus, while the interviewee’s perspective reflects an ideal scenario with abundant resources, the reality faced by the UTFPR demonstrates that the absence of a medical school presents practical challenges. The reliance on external providers can limit the development of a fully integrated and responsive occupational health system. This duality of perspectives highlights the complexity of implementing health policies in an environment that lacks specific internal resources, and both views contribute to a deeper understanding of the situation.

### 4.4. Action Recommendations

Based on document analysis and interviews, the following recommendations are made to improve SIASS implementation at the UTFPR:Allocation of dedicated professionals: The UTFPR should prioritize hiring specialized occupational health professionals, such as safety engineers and occupational physicians, to meet the growing demand. This planning should occur within the opportunities for opening new positions and be aligned with institutional demands;Budget allocation for preventive actions: In addition to reactive and mandatory actions, the UTFPR needs to allocate a portion of the annual budget to preventive initiatives promoting health and well-being. Long-term preventive policies should be prioritized over short-term projects;Creation of a continuous evaluation system: The UTFPR should create a formal system for evaluating occupational health policies, allowing the measurement of the impact of preventive actions and the adjustment of strategies as necessary. This continuous evaluation would ensure that initiatives are sustainable and effective in the long term.

In a transition scenario between the current situation and what is necessary to meet occupational health demands fully, simplifying bureaucratic processes within legal limits is essential to expedite the implementation of measures. Public–private partnerships (PPP) can also be a viable interim solution to address internal resource limitations. Additionally, developing bidding processes that avoid desert outcomes and ensuring the hiring of qualified suppliers can help overcome immediate challenges. These interim strategies will allow the UTFPR to move forward while internally developing the ideal capacity to implement occupational health policies efficiently.

### 4.5. Lessons Learned

Studies on worker health in Latin America indicate that the lack of awareness and institutional weakness, coupled with high exposure to occupational risks, are critical challenges in the region. Iunes [44] highlights that these factors result in little attention being given to occupational safety. Similarly, Mendonça et al. [15] discuss how bureaucracy and job insecurity hinder the effective implementation of worker health policies. Comparing the UTFPR with other regional institutions, it becomes clear that these challenges are shared continentally, reinforcing the need for a more robust approach to overcome administrative hurdles and ensure safer working conditions.

The analysis of this study on the implementation of the SIASS at the UTFPR offers several lessons that can be applied to other institutions and contexts:The importance of consistent implementation: The inconsistent implementation of occupational health policies, as demonstrated at the UTFPR, limits the positive impact that these policies can have on employee health. This fact demonstrates that leadership commitment and adequate resource planning are essential to translating policy guidelines into concrete actions;Adequate resource allocation: The shortage of human and financial resources significantly compromises the effectiveness of health policies. The lesson learned here is that budget planning must consider the specific needs of each institution so that these policies are long-term sustainable;The adoption of a proactive approach: Excessive emphasis on corrective rather than preventive measures reduces the impact of occupational health policies. The UTFPR and other institutions would benefit from a more proactive approach, focusing on preventive actions to avoid long-term health problems;A need for interinstitutional collaboration: Partnerships with other institutions, such as the one between the UTFPR and IFPR, were beneficial, but their discontinuation revealed vulnerabilities. Interinstitutional partnerships can be an effective solution to share resources and optimize the implementation of health policies;Continuous policy evaluation: The absence of a formal continuous evaluation system limits the ability to adapt health policies. Implementing a monitoring system is crucial to ensure that policies are adjusted as the needs of employees change over time.

### 4.6. Study Limitations

Although this study presents essential findings on implementing the SIASS and managing health and well-being at the UTFPR, it is essential to recognize its methodological limitations. The limited sample, consisting of only two interviewed managers, may not capture all the nuances of the challenges faced by the institution and may be insufficient to fully reflect employees’ perspectives on these issues thoroughly. Additionally, the scarce available records limit a more detailed analysis of the processes. However, these interviews complement the document analysis, enabling data triangulation and offering a broader view of institutional issues, partially mitigating the impact of the small sample size. Nonetheless, the qualitative nature of the data and the limited number of interviewees remain limitations that should be considered when interpreting the results.

Moreover, the exclusive focus on a single university without a medical school restricts the ability to generalize the findings to other institutions, especially those with different structures and resources, such as universities with medical schools. These institutions may face distinct challenges in implementing health and well-being policies, which limits the direct applicability of the results of this study to such contexts.

The analysis focused on current policies. As such, it may have overlooked emerging practices or innovative solutions implemented in other institutions that could provide additional examples of best practices to improve SIASS implementation. Despite these limitations, the study contributes to a deeper understanding of health management practices in federal educational institutions, providing a solid basis for future comparative investigations.

## 5. Conclusions

This study provides an in-depth view of implementing the SIASS and managing health and well-being at the UTFPR, highlighting significant gaps that affect its effectiveness. Although the SIASS was created to ensure safer and healthier work environments for public servants, its implementation at the UTFPR faces persistent challenges, such as resource shortages and bureaucratic hurdles. As the current predominantly reactive measures demonstrate, these facts hinder occupational health policies’ proactive and preventive nature.

A deeper reflection on these findings demonstrates the importance of institutional commitment and the proper allocation of resources to realize the SIASS’s objectives fully. A proactive approach, focused on preventive rather than corrective measures, is necessary to maximize the program’s potential. Collaboration with other institutions, such as the partnership between the UTFPR and IFPR, emerges as a critical solution for resource sharing and improving occupational health policies between institutions.

Future research needs to focus on specific gaps in the literature, such as the long-term effectiveness of interinstitutional collaborations, the role of preventive health measures in public institutions, and the impact of regulatory frameworks on the health and well-being of public servants in resource-limited settings. These unexplored areas are crucial for enhancing the design of health policies, especially in public sector institutions facing budget constraints.

While the study highlights the current implementation limitations, it also points to broader systemic issues, such as bureaucracy and the prioritization of core mission areas, that affect the overall well-being of employees. By addressing these challenges, institutions like the UTFPR can improve workplace health outcomes, set a standard for public health policies in Brazil, and provide a reference for other countries.

## Figures and Tables

**Table 2 ijerph-21-01341-t002:** Summary of interviewees’ responses on health and well-being at UTFPR.

Category	Summary of Interviewee Responses
Inspection and Prevention	How are inspections carried out in the work environments at the UTFPR to identify situations that are potentially harmful to the health and well-being of employees? … Can you describe the process and frequency of these inspections? … What are the main challenges faced during these inspections? E1: The assessments are requested by employees, management, or coordination and carried out by the occupational safety engineer, following a schedule according to the order of the requests. Since the institution has limited resources, occupational health companies are contracted through bidding to serve all campuses. The assessments are carried out whenever new environments appear, or those already assessed are modified. Access is not problematic since those responsible for the environments are notified in advance. The institution provides the equipment necessary for the measurements, ensuring the inspections are carried out without significant obstacles [42]. E2: The assessments are carried out on demand. Since the institution has only one professional in the area and the requests cover all campuses, some more than 600 km from the headquarters, service is often unfeasible. To meet this demand, specialized companies are put out to tender, but due to current legislation, the process is expensive, bureaucratic, and often unsuccessful. The main challenge faced is the need for more personnel [43].
	What are the main working conditions at the UTFPR that are considered harmful to the health and well-being of employees? … How are these conditions identified and studied? … Is there a specific protocol for dealing with these situations? E1: Activities are carried out in laboratories in chemistry, mechanics, electronics, food, and radiology. Thid also includes the electrical and hydraulic maintenance sectors and some of the school farms on the Dois Vizinhos Campus. The working conditions in these environments are identified and studied through a technical report issued by a qualified professional, such as the UTFPR occupational safety engineer or a company contracted through a bidding process. According to legislation, all the environments assessed as having risk have a risk map posted, including safety procedures, personal protective equipment (PPE) and collective protective equipment (CPE), and the measures to be adopted during an accident. A protocol makes personal protective equipment available, and collective protective equipment is installed in the necessary locations [42]. E2: The UTFPR has more than 1,000 laboratories, many of which have working conditions considered hazardous by law. The institution complies with current legislation to the best of its ability [43].
	What preventive or corrective measures have been implemented to eliminate or neutralize existing risks? … Can you give examples of measures that have been effective? E1: The installation of fume hoods, and protective devices on machinery, and the storage of hazardous materials in separate locations appropriate to the risk, such as gas cylinders and chemical inputs, and the provision of PPE [42]. E2: From the perspective of senior management, there is a continuous effort to allocate resources and provide bureaucratic support to meet existing demands [43].
Research and Monitoring	How is the investigation into the causes and consequences of work-related accidents and illnesses conducted at the UTFPR? … What are the main factors that contribute to these incidents? E1: Accidents at work are investigated through Administrative Proceedings to determine the causes and forward actions. The illnesses are assessed by medical experts and medical boards. The number of serious accidents is very small, each with its characteristics. No type of accident repeats itself successively [42]. E2: When an accident occurs, the investigation follows the legal procedures. Problems related to mental health are predominant but are, to some extent, for investigation outside the scope of the law [43].
	How are corrective measures monitored until their completion? … Can you describe the monitoring process and the challenges involved? E1: Monitoring is carried out by a UTFPR safety engineer and the responsible department. The biggest challenge is the lack of sufficient professionals in safety engineering. After the report is issued, the implementation of the measures is verified by the administration and planning department [42]. E2: Monitoring is, or should be, carried out by the responsible department. The institution’s management is aware of the lack of employees in the department, but has very limited possibilities to solve the problem, as it does not have the autonomy to hire new employees [43].
Education and Health Promotion	What are the leading occupational health and safety standards published at the UTFPR? … How is compliance with these standards promoted among employees? E1: Besides legislation, internal regulations, and service orders are disseminated in various ways and available on the institution’s website. Guidance is provided by the person responsible for the sector or activity through lectures and by sending out notices [42]. E2: Through the bodies responsible for the sectors, which must comply with the regulatory framework [43].
	Can you describe the educational campaigns for health, accident, and disease prevention that have been carried out at the UTFPR? … What were the results observed from these campaigns? … How have employees responded to these initiatives? E1: Courses and lectures are held annually during employee week, in which this information is included. Events of this exact nature may occur on other occasions. Posters and warnings are posted frequently. In addition, there is an Employee Development Plan, which refers employees to training whenever requested [42]. E2: The UTFPR is the largest university in the federal system in terms of number of campuses. In such an extensive system with significant differences between campuses, there is no uniformity, and no formal evaluations of results are carried out. However, success stories are often shared with other campuses. Staff have responded to these initiatives positively, understanding that they are also responsible for security and actively collaborating [43].
Negotiation and Improvement	How are negotiations conducted with managers to improve processes and work environments at the UTFPR? … Can you provide examples of improvements implemented due to these negotiations? E1: Based on the report, the human resources area is informed and coordinates the actions with the responsible sector and the administration. Both work in partnership to implement the measures that the Internal Audit monitors. Improvements include replacing floors, installing chapels, and replacing equipment [42]. E2: Negotiations follow the institution’s internal regulations and current legislation [43].
Impact of the Absence of a Medical Course	How does the absence of a Medicine course at the UTFPR impact institutional priorities in allocating resources for occupational health programs? … Due to this absence, are there any specific human or material resource difficulties? … How does this influence decisions about the occupational health of employees? E1: The absence of a Medicine course at the UTFPR does not directly affect the allocation of resources for occupational health since examinations and expert reports require specialist doctors, according to NR-7. The lack of Occupational Safety engineers on all campuses and in the rectory makes updating and monitoring occupational reports difficult. Although the SIASS allows medical expert reports in other agencies, hiring specialized companies through bidding can cause delays. The use of technology for remote expert reports has helped to mitigate these challenges [42]. E2: The existence of a Medicine course, in itself, does not solve the problem. However, even with possible functional deviations, institutions that offer this course face fewer problems in the federal system than those that do not. Although federal institutions have autonomy, they cannot hire professionals without authorization from the federal government, which makes this limitation a complex problem [43].
	In your opinion, how could the presence of a medical course change occupational health policies and practices at the UTFPR? … Do you believe there would be significant changes in the allocation of resources or the implementation of programs? E1: The existence of a medical course would not bring significant changes in the allocation of resources or the implementation of occupational health programs. This would increase the environments and servers to be monitored, while professionals would focus on teaching activities, not occupational medicine. Specific positions for doctors dedicated full-time to occupational medicine would be necessary to meet the sector’s needs. The occupational physician, linked to the SIASS, operates within a national system with defined regulations, and the transfer of funds to the university follows budgetary and public accounting standards [42]. E2: At the UTFPR, the possibility of implementing a medical course does not exist, mainly due to the institutional focus on technology and engineering courses. Therefore, it is not appropriate to consider how this could be accomplished [43].

Source: own authorship.

## Data Availability

Dataset available on request from the authors.

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
