# Peer review of "Shaping Public Servant Well-Being: Lessons from Brazil’s SIASS Program"

_ijerph, 2024, doi:10.3390/ijerph21101341_

Round 1

Reviewer 1 Report (Previous Reviewer 2)

Comments and Suggestions for Authors

Author Response

Comment

Response

General comments

1) Author Contributions: The authors should be abbreviated with the first letters of their names (see template).

Corrected.

General comments

2) Funding: Describe this in more detail. What exactly was funded. I see a conflict of interest here, as the funding can certainly influence the results.

The funding information has been corrected. We mistakenly considered the researchers' salaries as part of the funding. However, the study received no internal or external financial support for its execution.

General comments

3) Institutional Review Board Statement: When did the ethics committee vote?

Information is allocated in the Materials and Methods section.

Abstract

4) In summary, the abstract is well structured and provides a clear overview of the objectives and results of the study. However, it could be improved with more detailed methodological information and more specific recommendations.

The abstract has been revised to include more detailed methodological information, particularly regarding the type of data collected and the analytical methods used. Additionally, more specific recommendations have been added to provide clearer guidance for practical implementation.

Introduction

5) Abbreviations must be explained the first time they are used (Line 70). What is SIASS? What does it mean written out?

Added at the first mention in the text.

Introduction

6) The introduction (point 1) leans a little too heavily on the political context. The extent of the political discussion can easily be seen as a distraction from the actual scientific investigation. This weakens the manuscript as it keeps the reader in the general political discourse for too long before moving on to the actual results and methodology. A more concise introduction focused on the research topic could more clearly emphasize the scientific depth and relevance of the work.

We have revised the introduction to remove the excessive political discussion and reoriented it to focus on the health and well-being of public servants, specifically within the context of the SIASS program and its relevance to the research. The revised introduction now emphasizes the scientific investigation, the study’s objectives, and its relevance to public health, ensuring a clearer focus on the research topic and scientific depth.

Introduction

7) Introduction point 2 should be rewritten in poin 1.1.

Request fulfilled. The introduction has been restructured according to the referee's suggestions.

Introduction

8) The study objectives are similarly presented in lines 114-125 and again in lines 150-157. Both sections emphasize that the SIASS system aims to promote the health and well-being of public servants and mention challenges such as limited resources and uneven implementation. This repetition seems redundant and does not add anything new to the reader's understanding. Also, formulate the research questions for your study. It is best to include a section 1.2 Derivation of the research question for this study.

Request fulfilled. The introduction has been restructured according to the referee's suggestions.

Material and methods

9) Paragraph 3 should be paragraph 2. Adjust also the other sections.

Corrected.

Material and methods

10) Paragraph 3.1: Please explain why universities without medical faculties were selected? You already did that in the introduction. But it should be explained here. Please adapt this.

The comment has been addressed. The explanation about the selection of universities without medical schools was included in Section 2.1.

Material and methods

11) From which year do the evaluated documents originate?

Information allocated in the Materials and Methods section.

Material and methods

12) While there is mention of conducting a document analysis, there is no detailed description of the specific documents that were analyzed and how they were integrated into the study. It remains unclear what type of institutional documents were used and whether they were systematically analyzed or merely used to support the interviews.

Information allocated in the Materials and Methods section.

Material and methods

13) Although the case study methodology is useful, the small number of only two interviewees could be interpreted as a weakness. Two interviews provide only a limited perspective on health and wellbeing policy across the university. A larger number of interviewees representing different positions and perspectives within the institution would have provided a richer and more balanced set of data.

After discussing the point raised, the authors agree with the referee's observation. Upon reviewing the technical procedures employed, we have reclassified the study as documental research. This approach better aligns with the primary focus of the study, which relied on the systematic analysis of institutional documents. While the interviews provided valuable insights, they were supplementary to the documental analysis and not the central source of data. The section has been revised according to the highlighted observations to reflect this methodological procedure adjustment.

Material and methods

14) The section does not provide detailed insight into the analysis of the interviews. It only mentions that the interviews were transcribed and coded using NVivo, but it lacks specific information on how the data were coded, categorized, and interpreted. More details about the data analysis would increase methodological transparency and comprehensibility.

The section was expanded with details of the analysis of the interviews, adding information on how the data were coded, categorized, and interpreted, aiming to increase methodological transparency and comprehensibility.

Material and methods

15) Although it is explained why UTFPR was chosen as the subject of the study, it remains unclear whether other institutions were considered or why no comparison cases were included. A comparison with another university, possibly a medical school, would have added depth and context to the study.

The study focused on UTFPR, which represents the reality of most Brazilian federal institutions without medical schools. The inclusion of other institutions, such as medical schools, was not considered because the objective was to analyze a more common and relevant context for most of these institutions. The justification was included in the Materials and Methods section.

Results

16) While Table 2 provides a good overview of the UTFPR's annual health and wellbeing initiatives, the deeper analysis of this data remains relatively superficial. While it mentions that there have been initiatives, it lacks a more detailed analysis of how effective these measures have been and how to assess progress over the years.

The institutional reports do not provide detailed data on these initiatives' results or effectiveness over time, which limits a deeper analysis. This point has been clarified in the Results section, highlighting the need for greater institutional monitoring to assess the impact of these actions. Additionally, we expanded the description of the relationship between these data and the data extracted from the interviews, highlighting that current measures are predominantly reactive, and that there is a need for proactive and preventive measures.

Results

17) The qualitative data from the interviews are mentioned, but the analysis remains at a rather general level. There is a lack of detailed insight into how the interviewees' statements illustrate specific problems or successes, or how these relate to the quantitative data.

We have revised the analysis of the qualitative data from the interviews to provide a more detailed view of how the respondents' statements illustrate specific problems or successes. We included more concrete examples, highlighting the connection between the managers' perceptions and the challenges in implementing health and well-being policies.

Results

18) Although some challenges, such as the lack of resources and qualified personnel, are addressed, concrete solutions or recommendations for overcoming these problems are largely absent. A stronger focus on possible actions that could be derived from the findings would have been helpful.

The discussion has been revised to include concrete solutions and recommendations for overcoming the identified challenges. These include improving resource allocation, implementing better recruitment and training strategies for qualified personnel, and fostering collaboration between institutions. Specific actions based on the study’s findings are now provided to offer practical steps for improving health and well-being policies.

Discussion

19) A major weakness of the discussion is the lack of an explicit discussion of limitations. It is not addressed that the results may have been influenced by the small number of interviews and the limited sample. Methodological limitations (e.g. the sole focus on one university) are also not addressed. However, a discussion of limitations is essential to critically reflect on the validity and generalizability of the findings.

We revised the Discussion section to include a clearer analysis of the study's methodological limitations. We acknowledge that the small number of interviews and limited sample may have influenced the results, particularly in capturing all the nuances of institutional challenges and the employees' perspectives. Additionally, we reflected on the impact of focusing on a single technological university without a medical school, which may limit the generalization of the findings to other institutions.

These limitations are now addressed in detail, allowing for a more critical assessment of the results' validity and applicability in other contexts. We believe these revisions improve the clarity and strength of the discussion, addressing the referee's concerns.

Discussion

20) In the results, the respondents say that the lack of a medical faculty has no relevant impact on the health care of the staff. It is argued that health care is provided by external specialists and that the lack of a medical faculty is not directly related to the scarcity of resources (lines 273-275). In the discussion, however, it is emphasized that the lack of a medical faculty is an additional challenge (lines 318- 323). This contradiction between the interviewees' statements and the interpretation in the discussion weakens the coherence of the argumentation. The authors should have emphasized the different perspectives more clearly, or at least explained why the discussion came to a different conclusion.

In light of this observation, a contextualized explanation was structured in the Discussion section.

Discussion

21) Although the discussion describes the problems and challenges, it does not offer concrete solutions. The authors could have provided specific recommendations or suggestions for action to show how the university can overcome the identified challenges (e.g., limited resources). This would have been particularly important for the practical implementation of the findings.

The discussion has been revised to include specific recommendations for overcoming the identified challenges, such as enhancing resource allocation, strengthening partnerships, and adopting a proactive health management approach. These actions are now linked to practical steps that UTFPR and similar institutions can take to improve the implementation of SIASS.

Discussion

22) The discussion remains at a general level, mainly discussing structural and organizational challenges. However, there is a lack of in-depth analysis of the specific findings of the interviews, such as how the responses of the managers interviewed specifically reflect the problems identified in the literature. Greater integration of the qualitative data into the discussion would have added depth to the analysis.

We reorganized the qualitative interview data into a more structured discussion. We expanded the use of these qualitative data by linking them to the data extracted from the documents. Managers’ responses are now more clearly linked to the specific challenges identified in the literature, with greater integration of qualitative findings to provide a more complete analysis of how these issues affect the implementation of SIASS at UTFPR.

Discussion

23) A discussion organized into sections on document analysis, interviews, and recommendations for action would more clearly present the findings and better guide the reader through the different levels of analysis.

The discussion has been reorganized to include distinct sections for document analysis, interviews, and recommendations for action. This structure provides a clearer presentation of the findings and allows readers to follow the different levels of analysis more easily.

Discussion

24) The statements of the interviewed managers could be analysed in more detail and compared with the results of the document analysis.

We have expanded the comparison between the interview statements and the document analysis results. This comparative approach now highlights consistencies and differences between the qualitative and documental data, providing a more comprehensive understanding of SIASS implementation's challenges.

Discussion

25) Many South American countries have a similar political and institutional structure, which makes the comparison in the discussion more meaningful. This includes aspects such as the central role of the state in health care and the provision of public services. As far as this is possible from the literature, otherwise a brief mention in the discussion would be useful.

A comparative mention of South American countries' political and institutional structures has been added to the discussion. This brief comparison provides context for understanding how the State's central role in health care and public service provision may impact SIASS implementation at UTFPR and similar institutions.

Conclusions

26) The section reads more like a repetition of the results and discussion without providing any deeper reflection or new insights. The conclusions do not seem to provide any independent new insights that go beyond what has already been explained in the previous sections.

The conclusion has been rewritten to go beyond a summary of the results. It now includes a more reflective synthesis of the findings, highlighting overarching insights into the challenges of SIASS implementation and providing recommendations for improving occupational health management.

Conclusions

Conclusions should go beyond a simple summary and attempt to formulate more in-depth findings. What is missing is a synthesis of the findings that takes the reader to a new level and highlights the key overarching lessons of the study.

The conclusion has been revised to emphasize the key lessons learned from the study. It synthesizes the main findings, demonstrating the importance of institutional commitment, resource allocation, and interinstitutional collaboration to improve health and well-being policies.

Conclusions

Although the need for future research is emphasized, there is a lack of in-depth discussion of what specific research gaps exist in the current literature. It would have been useful to be more specific about which aspects of health and well-being policy in public institutions remain under-researched and how future studies could fill these gaps.

The conclusion now includes a more detailed discussion of the research gaps identified in the literature. It outlines specific areas for future research, such as the long-term effectiveness of partnerships between institutions, the impact of regulatory frameworks, and the role of preventive health measures in resource-constrained settings.

Reviewer 2 Report (New Reviewer)

Comments and Suggestions for Authors

The purpose of the paper is to identify and describe how the health and well-being of public servants are treated at the Federal University of Technology – Parana in Brazil. The study has a qualitative character – data is taken from documents and 2 interviews with persons responsible for implementation of The Integrated Subsystem for Attention to the Health of the Server at this university. The study has strong limitations (mentioned by the Authors) as it presents a single-case study which cannot be generalized. However, some important problems, advances and challenges are identified.

I have some particular remarks.

1.        I find that the title of the paper does not reflect the content properly. The paper does not describe health or well-being of servants but the management and promotion of them with respect to a particular federal initiative.

2.        In lines 195-213 you present the five phases of the analysis. You write that “Data from various sources, including documentation and interviews, were compiled and organized into a structured database”. And “The data were then coded and reorganized into predefined categories: Inspection and Prevention, Investigation and Follow-up, Health Education and Promotion, and Negotiation and Improvement.” But when presenting the results in Table 3 which seems to be crucial in the paper and includes these predefined categories, you mention only the materials from interviews and no other documentation. So in my opinion there is a discrepancy between sections 3.3 Data Analysis: A Five-Phase Approach and 4. Results  that needs to be clarified.

3.        In lines 203-204 you mention that “These categories were structured in alignment with the research questions.” I couldn’t find a direct formulation of research questions in your paper. In my opinion it would be better to formulate research questions and present them, perhaps in the Introduction.

Author Response

Comment

Response

I think the title of the article does not adequately reflect the content. The article does not describe the health or well-being of the public servants, but rather the management and promotion of these aspects in relation to a specific federal initiative.

The title has been changed to: Shaping Public Servant Well-being: Lessons from Brazil’s SIASS Program

In lines 195-213, you present the five phases of the analysis. You write that “Data from various sources, including documentation and interviews, were compiled and organized into a structured database.” And “The data were then coded and reorganized into predefined categories: Inspection and Prevention, Investigation and Monitoring, Health Education and Promotion, and Negotiation and Improvement.” However, when presenting the results in Table 3, which seems to be crucial to the article and includes these predefined categories, you only mention the interview materials and no other documentation. In my opinion, there is a discrepancy between sections 3.3 Data Analysis: A Five-Phase Approach and 4. Results, which needs clarification.

The predefined categories—Inspection and Prevention, Investigation and Monitoring, Health Education and Promotion, and Negotiation and Improvement—were derived from relevant legislation (SIASS guidelines) and used for the document and interview analyses.

While Table 3 emphasizes interview data, institutional documents were systematically analyzed to provide a comprehensive understanding of UTFPR's health and well-being policies. The documents allowed for data triangulation and supported the conclusions by cross-referencing institutional practices with interview insights.

We have revised the text in Sections 3.3 and 4 to clarify the integration of document analysis and interviews, ensuring both sources are properly reflected in the results.

In lines 203-204, you mention that “These categories were structured in alignment with the research questions.” I could not find a direct formulation of the research questions in your article. In my opinion, it would be better to formulate the research questions and present them, perhaps in the Introduction.

The research questions were presented in the Introduction.

Reviewer 3 Report (New Reviewer)

Comments and Suggestions for Authors

Thank you for the opportunity to review this article.

I think the methodology of the paper is missing. The qualitative methodology has clear steps, and I don´t see how the process of interviewing was done. Also, the model presented has no references.

The paper most be improved in order to be presented again.

Author Response

Comment

Response

I think the methodology of the article is lacking. The qualitative methodology involves clear steps, and I don't see how the interview process was conducted. Additionally, the presented model does not include references.

The Materials and Methods section was significantly expanded. The changes made address the concerns raised.

Round 2

Reviewer 1 Report (Previous Reviewer 2)

Comments and Suggestions for Authors

Thank you very much for the revision of the manuscript. 

Reviewer 3 Report (New Reviewer)

Comments and Suggestions for Authors

There are some minor errors of redaction in the new text

The new paper is much better than the old one, it meets the requirements of qualitative methodology

This manuscript is a resubmission of an earlier submission. The following is a list of the peer review reports and author responses from that submission.

Round 1

Reviewer 1 Report

Comments and Suggestions for Authors

In the manuscript, the authors have chosen the Federal Technological University of Paraná (UTFPR) as the subject of a single-case study to investigate the health and well-being of Brazilian civil servants. However, the rationale for selecting UTFPR is not clearly articulated. It is imperative that the authors provide a justification for this choice, detailing whether UTFPR was selected due to its representativeness, unique attributes, size, or other distinctive factors. This clarification is essential for assessing the generalizability of the findings and understanding any limitations the specific case may impose on the broader applicability of the study's conclusions.

The study, while emphasizing the importance of the workplace environment in influencing health and well-being, seems to focus more on policy support than on the actual conditions and dynamics of the workplace. The physical environment, work culture, employee relations, and job design are critical components that directly affect employees' health and well-being. The authors are encouraged to broaden their analysis to encompass these specific aspects of the workplace environment and to discuss how they interplay with policy support to foster a conducive environment for health and well-being.

Moreover, the manuscript reports the use of semi-structured interviews as a primary method for data collection but does not reflect the findings or insights from these interviews in the conclusion section. The qualitative data obtained from interviews can offer valuable perspectives on the research question. It is recommended that the authors include a summary of the interview findings in the conclusion to demonstrate how these qualitative insights support or extend the research findings, thereby enhancing the transparency and depth of the study.

Additionally, the termination of the partnership between UTFPR and IFPR in 2024 is mentioned, but the reasons behind this decision are not provided. Understanding the rationale for ending this partnership is crucial for evaluating its impact on the health services provided to UTFPR employees and the institution's ability to meet their health needs in the absence of this collaboration. The authors should consider elucidating the reasons for the partnership's termination to provide a more comprehensive analysis.

In summary, the manuscript would benefit from a clear justification for the case study selection, an expanded analysis that includes the actual workplace environment, integration of interview data in the conclusions, and an explanation for the termination of the UTFPR-IFPR partnership. Addressing these points will strengthen the study's methodology, relevance, and applicability.

Reviewer 2 Report

Comments and Suggestions for Authors
